# Germline POT1 Deregulation Can Predispose to Myeloid Malignancies in Childhood

**DOI:** 10.3390/ijms222111572

**Published:** 2021-10-26

**Authors:** Pia Michler, Anne Schedel, Martha Witschas, Ulrike Anne Friedrich, Rabea Wagener, Juha Mehtonen, Triantafyllia Brozou, Maria Menzel, Carolin Walter, Dalileh Nabi, Glen Pearce, Miriam Erlacher, Gudrun Göhring, Martin Dugas, Merja Heinäniemi, Arndt Borkhardt, Friedrich Stölzel, Julia Hauer, Franziska Auer

**Affiliations:** 1Pediatric Hematology and Oncology, Department of Pediatrics, University Hospital “Carl Gustav Carus”, TU Dresden, 01307 Dresden, Germany; Pia.Michler@uniklinikum-dresden.de (P.M.); Anne.Schedel@uniklinikum-dresden.de (A.S.); Martha.Witschas@uniklinikum-dresden.de (M.W.); UlrikeAnne.Friedrich@uniklinikum-dresden.de (U.A.F.); Maria.Menzel@uniklinikum-dresden.de (M.M.); 2Department of Pediatric Oncology, Hematology and Clinical Immunology, Medical Faculty, Heinrich-Heine University Duesseldorf, 40225 Duesseldorf, Germany; Rabea.Wagener@med.uni-duesseldorf.de (R.W.); Triantafyllia.Brozou@med.uni-duesseldorf.de (T.B.); Arndt.Borkhardt@med.uni-duesseldorf.de (A.B.); 3Institute of Biomedicine, School of Medicine, University of Eastern Finland, Yliopistonranta 1, FI-70211 Kuopio, Finland; juha.mehtonen@uef.fi (J.M.); merja.heinaniemi@uef.fi (M.H.); 4Institute of Medical Informatics, University of Muenster, 48149 Muenster, Germany; carolin.walter@uni-muenster.de; 5Department of Neuropediatrics Charité-Universitätsmedizin Berlin, Corporate Member of Freie Universität Berlin, Humboldt-Universität zu Berlin, and Berlin Institute of Health, 10117 Berlin, Germany; Dalileh.nabi@charite.de; 6Institute of Physiological Chemistry, Medical Faculty “Carl Gustav Carus”, TU Dresden, 01307 Dresden, Germany; glen.pearce@tu-dresden.de; 7Department of Pediatrics and Adolescent Medicine, Division of Pediatric Hematology and Oncology, Faculty of Medicine, University Medical Center Freiburg, 79106 Freiburg, Germany; miriam.erlacher@uniklinik-freiburg.de; 8German Cancer Consortium (DKTK), 79106 Freiburg, Germany; 9German Cancer Research Center (DKFZ), 69120 Heidelberg, Germany; 10Department of Human Genetics, Hannover Medical School, 30625 Hannover, Germany; Goehring.Gudrun@mh-hannover.de; 11Institute of Medical Informatics, Heidelberg University Hospital, 69120 Heidelberg, Germany; martin.dugas@med.uni-heidelberg.de; 12Hematology and Oncology, University Hospital “Carl Gustav Carus”, TU Dresden, 01307 Dresden, Germany; Friedrich.Stoelzel@uniklinikum-dresden.de; 13National Center for Tumor Diseases (NCT), 01307 Dresden, Germany; Franziska.Auer@uniklinikum-dresden.de

**Keywords:** acute myeloid leukemia, pediatric, trio sequencing, germline cancer predisposition, POT1, shelterin complex

## Abstract

While the shelterin complex guards and coordinates the mechanism of telomere regulation, deregulation of this process is tightly linked to malignant transformation and cancer. Here, we present the novel finding of a germline stop-gain variant (p.Q199*) in the shelterin complex gene *POT1*, which was identified in a child with acute myeloid leukemia. We show that the cells overexpressing the mutated *POT1* display increased DNA damage and chromosomal instabilities compared to the wildtype counterpart. Protein and mRNA expression analyses in the primary patient cells further confirm that, physiologically, the variant leads to a nonfunctional *POT1* allele in the patient. Subsequent telomere length measurements in the primary cells carrying heterozygous *POT1* p.Q199* as well as *POT1* knockdown AML cells revealed telomeric elongation as the main functional effect. These results show a connection between *POT1* p.Q199* and telomeric dysregulation and highlight *POT1* germline deficiency as a predisposition to myeloid malignancies in childhood.

## 1. Introduction

Telomeres play an essential role in preserving our genetic material by protecting chromosomal ends from degradation, while at the same time avoiding unwanted DNA damage responses [1]. The shelterin complex, which consists of six different protein subunits—TRF1, TRF2, RAP1, TIN2, TPP1 and POT1 (Figure 1A)—is responsible for safeguarding and properly maintaining telomeric DNA [2]. Shelterin deregulation leads to uncapped telomeres, which subsequently risks irreversible cellular changes, including genome instabilities, cellular aging or senescence and malignant transformation [3].

Protection of telomeres 1 (POT1), located on chromosome 7q, is an indispensable part of the shelterin complex [4,5]. In humans, POT1 uniquely recognises and binds telomeric single-stranded (ss) DNA via its N-terminal oligonucleotide/oligosaccharide-binding (OB) domains (OB1 and OB2) [6]. C-terminally, POT1 connects to the other shelterin complex proteins by interacting with TPP1 [7,8,9]. While POT1 exerts a multitude of functions, the most prominent comprise regulation of the telomerase-dependent telomere length [5,6], as well as repression of ATR-mediated DNA damage responses [10,11]. Therefore, it is not surprising that POT1 malfunction has been linked to the development of various types of human cancer [12]. While somatic *POT1* mutations are most prevalent in angiosarcoma [13], monoallelic *POT1* germline mutations are associated with a wide range of cancers, including familial melanoma, chronic lymphocytic leukemia, angiosarcoma and glioma [12,14]. Recently, germline *POT1* variants were found to predispose to myeloid and lymphoid neoplasms in adults [15]. Nevertheless, whether *POT1* germline aberrations can likewise confer susceptibilities to childhood cancer is unclear. Here, we present a novel loss of function stop-gain mutation in *POT1* (p.Q199*) in a boy with acute myeloid leukemia (AML). We further show how the resulting *POT1* haploinsufficiency confers telomere elongation and genomic instability, thereby generating a susceptible environment for malignant transformation.

## 2. Results

### 2.1. Identification of a Novel Germline POT1 Stop-Gain Mutation (p.Q199*) in a Child with AML

To elucidate whether shelterin complex mutations can predispose to the development of pediatric cancer, we analysed whole exome sequencing data of two independent parent–child cohorts of pediatric cancer patients (TRIO-D, *n* = 158 [16]; TRIO-DD, *n* = 111) for rare germline variants (minor allele frequency (MAF) < 0.2%) in the shelterin complex genes (Appendix A). Overall, 23 variants were identified in 269 pediatric cancer patients (Figure 1B) across various tumor entities (Figure 1C), with missense mutations being the most prominent. While one patient (case 35) presented with both a *TERF2* variant and a *TINF2* variant, all the other variants were mutually exclusive (Figure 1D).

Interestingly, a novel heterozygous *POT1* stop-gain mutation was found in an 8-year-old boy with AML. This germline variant (ENST00000357628.8:c.595C>T) is not yet described in public databases and causes the substitution of a glutamine to a stop codon at position 199 (p.Q199*) of the POT1 protein. Sanger sequencing validated the presence of the variant in the germline of the affected boy (variant allele frequency (VAF) of 0.5), while the variant was absent in his healthy father (Figure 1E, Appendix A). Since his mother’s death at the age of 43 was not cancer-related, it is not clear whether *POT1* p.Q199* arose de novo or was maternally transmitted. Apart from the *POT1* variant, no other pathogenic or likely pathogenic variants according to the ACMG guidelines were identified in this patient (Appendix A). Accordingly, the *POT1* germline variant might play a role in the AML onset in this child. Clinically, the boy itself presented with pancytopenia, aplastic bone marrow and 70% myeloid blast cells in the bone marrow at the age of 8. Furthermore, the somatic molecular genetic makeup of the myeloid blast cells displayed an atypical monosomy 7q- with derivative chromosome 7, potentially leading to a loss of the remaining *POT1* wild-type (WT) allele (located on chromosome 7q) in the tumor (Appendix A). Since the derivative chromosome 7 was still present in the blast cells, loss of heterozygosity could not be unambiguously verified in the tumor sample (detected VAF of 0.5). Due to an insufficient treatment response, the patient underwent hematopoietic stem cell transplantation with his haploidentical half-brother as the donor. Subsequently, the patient had a fast and sustained engraftment with a complete donor chimerism of 100% and is currently tapered from immunosuppression without any signs of graft-versus-host disease and relapse-free on day 526 after haploidentical cell transplantation.

### 2.2. POT1 p.Q199* Leads to a Loss of POT1 Expression from the Respective Allele

*POT1* p.Q199* is located within the oligonucleotide-/oligosaccharide-binding 2 (OB2) domain, subsequently leading to a loss of around 2/3 of the full-length POT1 protein, including its complete TPP1-binding domain. Furthermore, the novel *POT1* p.Q199* variant was not found in previous studies on germline *POT1* variants in adults with myeloid malignancies (lollipops in Figure 2A). The low frequency of *POT1* germline variants within the healthy population (gnomAD database V2.1, *n* = 118,479) and a constraint score of 0.18 indicating high intolerance of the protein towards loss-of-function variants (Figure 2A) strongly suggest a functionally relevant role of POT1 in cancer susceptibility.

To assess the immediate effect of *POT1* p.Q199* on gene expression, we performed quantitative real-time (qRT)-PCR analysis. Thereby, we could confirm haploinsufficiency of *POT1* in the patient’s fibroblast cells harboring heterozygous *POT1* p.Q199* compared to a *POT1* WT fibroblast control (*p* ≤ 0.0001, Student’s *t*-test; Figure 2B), as well as in the patient’s nonleukemic peripheral blood mononuclear cells referenced to blood from his father (*p* = 0.024, Student’s *t*-test; Figure 2C). Additionally, Western blot analyses of the patient’s primary fibroblasts revealed reduced POT1 levels, further validating the loss of one intact *POT1* allele in the patient (Figure 2D).

### 2.3. POT1 p.Q199* Overexpression Confers Increased DNA Damage Response Induction

Depletion of POT1 is known to cause de-repression of ATR-dependent DNA damage responses [10]. To test a potential functional influence of the truncated POT1 p.Q199* protein with regard to DNA damage signaling, N-terminally c-Myc-tagged *POT1* p.Q199* was cloned, transfected into HEK293T cells and compared to its WT counterpart (Appendix A). Western Blot analysis of the c-Myc-tag confirmed overexpression of the WT POT1 as well as a truncated protein corresponding to p.Q199* in the respective cell line system (Appendix A). Subsequent DNA damage analyses showed that POT1 p.Q199* overexpression led to an increase of DNA double-strand breaks in transfected HEK293T cells determined by gammaH2AX and 53BP1 immunofluorescence assays (*p* ≤ 0.001, Student’s *t*-test) (Figure 3A), which is in line with a deregulated DNA damage response and inappropriate repair by nonhomologous end joining [17]. These results could further be enhanced during stress conditions (irradiation with 3Gy, *p ≤* 0.001, Student’s *t*-test) (Figure 3B). Interestingly, compared to the overexpression system, the primary fibroblasts of the patient carrying heterozygous *POT1* Q199* did not corroborate this deregulated DNA damage phenotype (Figure 3C,D), suggesting another mode of action to enable malignant transformation in the physiological setting.

### 2.4. POT1 p.Q199* Leads to Telomere Elongation and Chromosomal Instability

Instead of a deregulated DNA damage phenotype, the patient’s fibroblasts showed dysregulation of another main POT1 function, namely telomere length maintenance. Here, relative telomere length measurements by means of qRT-PCR indicated significant telomere elongation in primary fibroblast cells harboring *POT1* p.Q199* compared to WT *POT1* control fibroblasts from an age-matched child (*p* = 0.019, Student’s *t*-test) (Figure 4A). This phenotype became even more apparent when the culture time was prolonged and the telomere length was assessed at a high passage number (passage 20: *p* ≤ 0.0001, Student’s *t*-test) (Figure 4A). The same trend of longer telomeres was observed in the HEK293T overexpression system, as well as in the patient’s blood cells (Appendix A). Elongated telomeres in the boy’s peripheral blood (granulocytes) could further be validated by clinical diagnostics (ΔΔCt value of 1.42). Corroborating these results, the second telomere length analysis of the patient’s blood after stem cell transplantation (haploidentical half-brother) confirmed reversion of the elongated telomere phenotype (Appendix A).

Additionally, to analyse chromosomal stability, telomere FISH assays on metaphase chromosomes were carried out. In line with our results on telomere deregulation, the *POT1* p.Q199* HEK293T cells showed a significant increase in telomere fragility compared to the cells overexpressing WT POT1 (*p* = 0.0002, Student’s *t*-test) (Figure 4B).

### 2.5. Reduced POT1 Levels in Myeloid Cells Confer Telomere Elongation

Within the hematopoietic system, POT1 shows the highest expression levels in hematopoietic stem and progenitor cells, as visualised by single-cell RNA sequencing data from the Human Cell Atlas (Figure 5A), while Western blot analyses show varying levels of POT1 in the ALL and AML cell lines (Appendix A). The clinical phenotype of the boy harboring the germline stop-gain *POT1* variant suggests a link between *POT1* p.Q199* and susceptibility to myeloid malignancies with 7q loss. To test this hypothesis, we generated a shRNA-mediated POT1 knockdown model of the AML cell line HL-60 (*POT1* mutation status = WT). Therefore, we used two different shRNAs—shRNA1 with a predicted knockdown level of 56% and shRNA2 with a predicted knockdown level of 92%. Successful transfection was validated via Sanger sequencing (Appendix A) and POT1 haploinsufficiency in both shRNA cell line models was confirmed by qRT-PCR (Figure 5B). Subsequent telomere length measurements confirmed significant telomeric elongation in the cells carrying the stronger knockdown with shRNA 2 (Figure 5C), thereby corroborating our previous results seen in the *POT1* p.Q199* overexpression model as well as in the primary patient cells.

## 3. Discussion

Telomeres act as “molecular clocks” by defining the proper lifespan of each cell. Accordingly, deregulation of the shelterin complex which guards and maintains telomeric DNA is closely linked to cancer progression [2,18]. Here, we present a pediatric patient with AML who harbors a novel stop-gain variant, conferring germline haploinsufficiency of the shelterin complex gene *POT1*.

We show that the HEK293T cells overexpressing POT1 p.Q199* display an increased number of gammaH2AX foci and chromosomal fragility compared to WT POT1 overexpression, which points towards a loss of function phenotype. This also corroborates the previous findings of the induction of DNA damage signaling and telomere fragility in *Pot1a* deficient murine models [19,20]. Compared to the overexpression model, the primary fibroblasts harboring heterozygous *POT1* p.Q199* displayed strong telomeric elongation. This is in line with published data from human embryonic stem cells (hESC), in which cancer-associated *POT1* mutations did not trigger DNA damage responses but led to longer telomeres [21]. This telomere elongation is further confirmed by numerous *in vitro* POT1 knockout and overexpression models [5,21,22,23].

Causal relationships between longer telomeres and an increased cancer risk have been reported [24]. Interestingly, even though AML is commonly associated with short telomere syndromes (STS) [25], our data support an opposite scenario of telomere elongation mediated by *POT1* p.Q199* in our patient. Although counterintuitive, this observation is in line with the absence of classical STS phenotypes [3] in our patient and adult AML patients harboring *POT1* variants [15]. Moreover, previous reports on *POT1* germline variants in other cancers connect *POT1* deregulation with telomere elongation [26,27]. Therefore, our findings suggest that germline *POT1* haploinsufficiency causes abnormally long telomeres, which might generate a susceptible cell population with extended proliferative capacity and the potential to acquire additional mutations required for malignant transformation [21]. Likewise, long telomeres are more fragile and pose an increased risk for genomic instabilities, favoring cancer progression [18]. *POT1* is mostly expressed in stem and progenitor cells as depicted by scRNA-Seq data and human hematopoietic stem cells (HSCs) harboring CRISPR-Cas9-induced heterozygous *POT1* stop-gain mutations do not display fitness disadvantages in vivo [21]. Therefore, the precursor cells of the hematopoietic system could be directly affected by *POT1* aberrations, which is in line with the clinical phenotype observed in the patient.

Taken together, our data highlight a potential role of *POT1* germline deregulation in the context of predisposition to myeloid malignancies in childhood, which is mediated through telomere elongation. Although still in its infancy [28], antitelomerase therapy should be considered as a potential anticancer strategy to counteract malignant transformation through longer telomeres in *POT1*-deficient tumors.

## 4. Materials and Methods

### 4.1. Patients

Patients ≤ 19 years of age were unselectively recruited at the Pediatric Oncology Department, Dresden (years 2019–2021), or as previously described [16]. Consent of the families was obtained according to Ethical Vote EK 181,042,019 (Dresden) and in line with the Declaration of Helsinki.

### 4.2. Whole Exome Sequencing (WES)

Germline DNA was extracted from the patients’ fibroblasts using an AllPrep DNA/RNA Mini Kit (Qiagen, Venlo, Netherlands) and from PBMCs of the parents and the remaining patients using a QIAamp DNA Blood Mini Kit (Qiagen). Next-generation libraries were generated with a SureSelect Human All Exon V7 kit (Agilent Technologies, Santa Clara, California). The libraries were sequenced on a NovaSeq 6000 platform (Illumina) in the paired-end mode (2 × 150 bp) and with the final target coverage of ≥100×. Read files in the fastq format were generated with bcl2fastq v2.19.0, and trimmomatic v0.33 was used to remove adapter and low-quality sequences [29]. The alignment to the human reference genome GRCh37 was performed using BWA-MEM v0.7.12 [30] and Samtools v1.2 [31]. The tool Peddy 0.4.6 [32] performed gender and relatedness analyses to validate the correct sample assignment and the expected relationship of the patient’s data with the corresponding parents’ data. Single nucleotide variants (SNVs) and insertion/deletions (indels) were called using GATK v4.1.4.1 and VarScan2 v2.3.9 [33], applying the trio mode.

Initial variant interpretation was carried out with the CPSR pipeline [34], which classified the variants as pathogenic, likely pathogenic, variant of unknown significance (VUS), likely benign or benign. Additional variant interpretation was manually performed (e.g., by taking the CADD scores into account [35] as well as by utilising an extended cancer gene list).

### 4.3. Sanger Sequencing Validation

*POT1* p.Q199* was validated via PCR and subsequent Sanger sequencing using the forward primer ACTCTACTCTCTTATGGCAGGT and the reverse primer CATCACCTTCAGAGATCTTGCC (5′ → 3′).

### 4.4. POT1 Variation Analysis

Allele frequencies of all the coding germline variants present in *POT1* (Ensembl transcript ID ENST00000297338) in a global healthy population taken from the gnomAD noncancer exome r.2.1.1 dataset (*n* = 118,479) were summed up codon-wise. The variants must have been VEP-annotated to one of the following consequences for inclusion: start_lost, missense_variant, inframe_insertion, inframe_deletion, stop_gain, frameshift_variant, coding_sequence_variant, stop_lost, incomplete_terminal_codon_variant, transcript_ablation, transcript_amplification, protein_altering_variant. The collected dataset was smoothed using the LOWESS algorithm (fraction: 0.06, iterations: 3) prior to plotting.

### 4.5. Cloning

The coding sequence of human WT POT1 was obtained from plasmid pLPC mycPOT1 (a gift from Titia de Lange (Addgene plasmid No. 12387; http://www.addgene.org/12387/) (accessed on 25 October 2021); RRID: Addgene 12387) [5]. The mutant sequence for POT1 p.Q199* was created using site-directed mutagenesis by PCR. After digestion of the PCR product and vector, ligation of the vector DNA and the insert DNA was performed using a Quick Ligation™ Kit (New England Biologies (NEB) No. M2200S, Ipswitch, Massachusetts). Recombinant DNA was introduced into 10-beta competent *E. coli* (NEB No. C3019H) by transformation. The organisms containing vector sequences were selected and validated by Sanger sequencing. The plasmids were amplified in maxi cultures and POT1 WT/Q199* DNA was purified with a NucleoBond^®^ Xtra Maxi EF Kit (Qiagen, Venlo, Netherlands) according to the manufacturer’s instructions.

### 4.6. Cell Culture

The primary fibroblasts were cultivated in the BIO-AMF™-2 Medium (Biological-Industries, Kibbutz Beit Haemek, Israel) for up to five passages. For experimental analyses, the fibroblasts were cultured in Dulbecco’s Modified Eagle’s Medium (DMEM, obtained from GIBCO/Thermo Fisher Scientific, Waltham, Massachusetts) with 20% fetal calf serum (FCS; GIBCO), 1% penicillin/streptomycin (P/S; 10,000 units/mL; GIBCO) and 1% MEM Non-essential Amino Acids (NEAA; GIBCO).

HEK293T cells were cultured in DMEM with 10% FCS, 1% P/S and 1% NEAA. HL-60 cells and the additionally tested leukemia/lymphoma cell lines were cultivated in the RPMI 1640 (GIBCO) medium with 20% FCS and 1% P/S. All the cells were kept at 37 °C and 5% CO_2_.

### 4.7. HEK293T Cell Transfection

HEK293T cells were seeded at a density of 4 × 10^5^ cells and transfected with POT1 WT and POT1 Q199* plasmids using Lipofectamine 2000 (Invitrogen/Thermo Fisher Scientific, Waltham, Massachusetts) according to the manufacturer’s instructions. The cells were selected with 2 µg/mL puromycin (Invitrogen), reduced to 1 µg/mL after one week. Validation of successful transfection was done by Sanger sequencing of reverse-transcribed mRNA.

### 4.8. POT1 Knockdown in HL-60 Cells

For lentivirus production, 15 × 10^6^ HEK293T cells were seeded on 15 cm dishes and transfected with VSV-G, pCMVdr8.2dvpr and shRNA plasmids containing specific POT1 knockdown sequences or non-coding shRNA. The transfection was performed using Lipofectamine 2000 according to the manufacturer’s instructions. The following plasmids were obtained from Sigma-Aldrich, St. Louis, Missouri: control (SHC016), shRNA1 (TRCN0000039804; predicted knockdown level, 0.56), shRNA2 (TRCN0000010352; predicted knockdown level, 0.92). The plasmids were amplified in maxi cultures and POT1 WT/Q199* DNA was purified using a NucleoBond^®^ Xtra Maxi EF kit (Qiagen).

HL-60 cells were transduced via spinfection on Retronectin (TakaraBio, Saint-Germain-en-Laye, France) coated 6-well plates. The successfully transduced HL-60 cells were selected with 1 µg/mL puromycin (Invitrogen), which was reduced to 0.5 µg/mL after 1 week. Knockdown of POT1 in HL-60 cells was quantified by qRT-PCR using POT1 TaqMan assays (Hs01565611_m1).

### 4.9. Quantitative Real-Time (qRT)-PCR Analysis

RNA was extracted from the primary fibroblasts (TRIO_DD_018; TRIO_DD_025; 2.0–3.0 × 10^6^ cells) using an RNaeasy Mini Kit (Qiagen No. 74106) with 350 µL of the RLT Buffer+ beta-ME using a QIAshredder (Qiagen, No. 79656) and an RNAse-Free DNase Set (Qiagen No. 79254). Three independent RNA extractions were performed. One µg of RNA was reverse-transcribed into cDNA using a QuantiTect Reverse Transcription Kit (Qiagen No. 205311) following the manufacturer’s instructions. Quantitative RT-PCR was performed using a TaqMan Universal Master Mix II following the manufacturer’s instructions (Thermo Fisher Scientific No. PN4428173, Waltham, Massachusetts) for a 20 µL reaction with 1.5 µL of cDNA. The following TaqMan assays (Thermo Fisher Scientific, Waltham, Massachusetts) were used: TBP (Hs00427620_m1), HPRT1 (Hs02800695_m1) and POT1 (Hs01565611_m1; Chr.7: 124,822,386–124,929,983). Expression of mRNA was analysed by means of the comparative ΔΔC_T_ method and plotted in relation to the control sample.

### 4.10. Western Blot

For whole cell lysates, 2–2.5 × 10^6^ (fibroblast samples) and 5 × 10^6^ (HEK293T cells stably overexpressing POT1 WT or p.Q199*) were lysed in a RIPA buffer (50 mM Tris, 150 mM NaCl, 0.5% sodium deoxycholate, 1% Triton and 0.1% SDS 20%, with 10× PhosSTOP (PS, Roche) and 25× PIC (Protease Inhibitor Cocktail, Roche, Basel, Switzerland) for 30 min on ice. The protein concentration was measured with the Bradford protein assay (Roti-Quant, Roth) by determining OD_595nm_. Twenty µg (HEK293T cells protein) or 15 µg (fibroblast cells protein) were heated for 10 min at 95 °C while shaking at 350 rpm and loaded accordingly onto a BIORAD Mini-Protean TGX Gel 4–20% (Bio-Rad Laboratories, Hercules, California). The blot was run for 3 h at 80 V. The transfer was performed using a Trans-Blot Turbo 1× Transfer System (high-molecular-weight, BIO-RAD). The immunoblot was blocked in 5% milk at room temperature for 1 h. After three washes with 1× TBS-T, the blot was incubated overnight at 4 °C with the following antibodies: a c-myc-Taq antibody (Invitrogen No. MA1-980, 1:1000) diluted in 5% Bovine Serum Albumin (Sigma-Aldrich, St. Louis, MO, USA) and POT1 (Novusbio No. NB500-176, 1:1000, Centennial, Colorado), GAPDH (Cell Signaling No. 5174S, 1:1000, Danvers, Massachusetts) diluted in 5% milk. The following day, the secondary antibody was applied after three consecutive washes (Cell Signaling Anti-Rabbit IgG No. 7075 1:1000, Cell Signaling Anti-Mouse IgG No. 7076) for 1 h in the dark, at room temperature, diluted in 5% milk. After three consecutive washes, the blot was imaged after the application of an HRP-linked solution (SuperSignal West Pico PLUS Chemiluminescent Substrate, Thermo Fisher Scientific, Waltham, MA, USA).

### 4.11. QRT-PCR of the Telomere Length (TL)

DNA was isolated from the fibroblast samples, the blood samples and the stably transfected HEK293T cells overexpressing POT1 using an AllPrep DNA/RNA/Protein Mini Kit (Qiagen) and stored at −20 °C. Afterwards, DNA was diluted to a concentration of approximately 25 ng/µL. The relative telomere length was measured by qPCR using a Relative Human Telomere Length Quantification qPCR Assay kit (Science Cell No. 8908) following the manufacturer’s instructions. For the HL-60 cells, the annealing time was extended to 30 s. Three biological replicates of each sample were analysed. For quantification of the TL, the comparative ΔΔC_T_ method was applied.

### 4.12. Telomeric FISH on Metaphase Chromosomes

One million POT1 WT and POT1 p.Q199* HEK293T cells were seeded in T75 flasks and incubated for 48 h at 37 °C and 5% CO_2._ Colchicine (Sigma-Aldrich) was added at a concentration of 5 µg/mL for 2 h. After trypsinisation, the cells were gently exposed to a hypotonic solution containing 0.075 M KCl and incubated for 10 min at 37 °C and 5% CO_2_. The cells were first fixed in methanol/acetic acid (3:1), placed on slides and stored at −20 °C. For performing fluorescence in situ hybridisation, Telomere PNA FISH Kit/Cy3 (Dako/Agilent Technologies No. K5326, Santa Clara, California) was used following the manufacturer’s instructions. The denaturation step was optimised to 90 °C for 10 min, and the hybridisation time was extended to 2 h at RT.

The slides were mounted in the ProLong Diamond Antifade mounting medium containing DAPI (Thermo Fisher Scientific) and stored at −20 °C. Metaphase spreads were captured with a Zeiss Axio Observer microscope using a Plan Apochromat objective with 63× magnificationaccompanying to the Core Facility Cellular Imaging (CFCI), Dresden, Germany). The DAPI images were used for featuring the metaphase chromosomes. At least 10 metaphase spreads per sample were captured and analysed. Chromosomal aberrations are presented as the frequency per metaphase.

### 4.13. Immunofluorescence Staining

Fibroblasts and stably transfected HEK293T cells overexpressing POT1 were plated onto Poly-L-Lysine pre-coated coverslips in 24-well-plates at a density of 2 × 10^4^ cells (fibroblasts) or 6 × 10^4^ cells (HEK293T) and cultured for 24 h at 37 °C and 5% CO_2_. The cells were exposed to 3 Gy (HEK293T cells) or 6 Gy (fibroblasts) ionising radiation and cultured again for 24 h as described above. The following day, the cells were fixed for 15 min in 3% formaldehyde/PBS, blocked with 0.25% Triton X-100/PBS for 10 min and blocked again in 1% bovine serum albumin/PBS for 30 min. The samples were incubated with the primary antibodies for 1 h at RT. Using a mouse polyclonal anti-phospho-histone H2AX (Ser139) antibody (Merck Millipore No. 05-636, Burlington, Massachusetts) at a dilution of 1:100 and a rabbit 53BP1 antibody (Novusbio No. NB100-304) at a dilution of 1:1000, yH2AX and 53BP1 foci were detected. Coverslips were further stained with the secondary antibodies for 1 h at room temperature in the dark. The goat anti-mouse Alexa Fluor 488 IgG antibody (Invitrogen No. A-11029) and the goat anti-rabbit Alexa Fluor 594 IgG antibody (Invitrogen No. A-11037) were used as the secondary antibodies, each at a dilution of 1:200. The slides were mounted in the ProLong Diamond Antifade medium containing DAPI. Wide-field microscopy was performed with a Zeiss Axio Observer microscope using a Plan Apochromat objective with 20×/40× magnification accompanying to the Core Facility Cellular Imaging (CFCI), Dresden, Germany). The DAPI images were used to detect signals inside the nuclei.

### 4.14. Single-Cell RNA Sequencing Analysis

Healthy human bone marrow scRNA-seq data from eight donors were downloaded from the Human Cell Atlas (https://data.humancellatlas.org/explore/projects/cc95ff89-2e68-4a08-a234-480eca21ce79) (accessed on 12 August 2021) [36] and aligned to hg19 using Cell Ranger v3.0.0. Scanpy (https://doi.org/10.1186/s13059-017-1382-0) (accessed on 12 August 2021) was used to characterise the cell types in the data, correcting for possible batch effects with the mutual nearest neighbors (https://doi.org/10.1038/nbt.4091) (accessed on 12 August 2021) and filtering for outliers using the median absolute deviation. The cell clusters found with Louvain clustering (https://zenodo.org/record/1054103) (accessed on 12 August 2021) were mapped to cell types using the known marker genes. Cell cycle phases were annotated by scoring cell cycle marker gene sets from https://doi.org/10.1101/gr.192237.115 (accessed on 12 August 2021). Two-dimensional visualisation was performed with UMAP (https://doi.org/10.1038/nbt.4314) (accessed on 12 August 2021).

### 4.15. Statistical Analyses

For the statistical analysis, two-tailed Student’s unpaired *t*-test was performed. Differences with a *p*-value < 0.05 were considered to be significant; ns = *p* > 0.05, * *p* ≤ 0.05, ** *p* ≤ 0.01, *** *p* ≤ 0.001, **** *p* ≤ 0.0001.

## Figures and Tables

**Figure 1 ijms-22-11572-f001:**
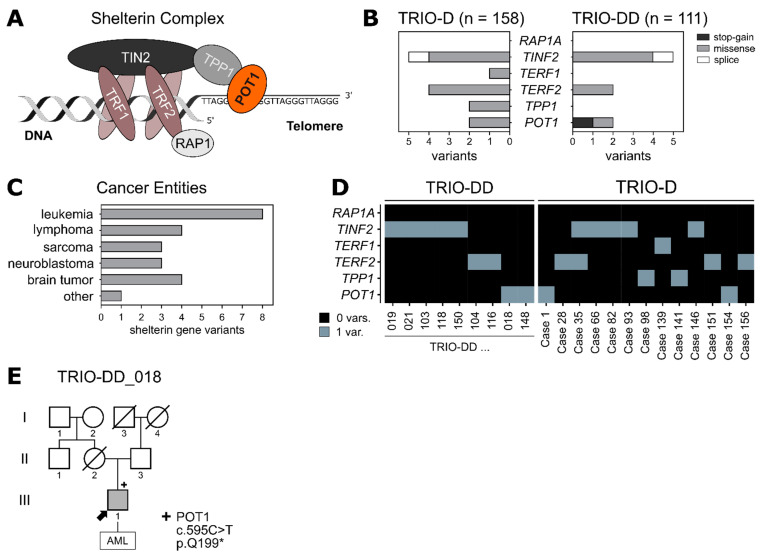
(**A**) Schematic drawing of the shelterin complex. POT1 (orange) is associated with the core complex via TPP1. (**B**) Two pediatric patient cohorts (TRIO-D, *n* = 158; TRIO-DD, *n* = 111) were analysed for germline variants within the shelterin complex genes. Only protein-altering variants with a MAF < 0.2% (healthy population) were included. (**C**) Tumor entities of the patients harboring the shelterin complex variants identified in (**B**). (**D**) Distribution of the protein-altering variants from (**B**) among the patients. (**E**) Family pedigree of the patient carrying the heterozygous germline *POT1* variant p.Q199*. The index patient is marked with an arrow. Variant carrier is marked with “+”. Half-siblings were not included.

**Figure 2 ijms-22-11572-f002:**
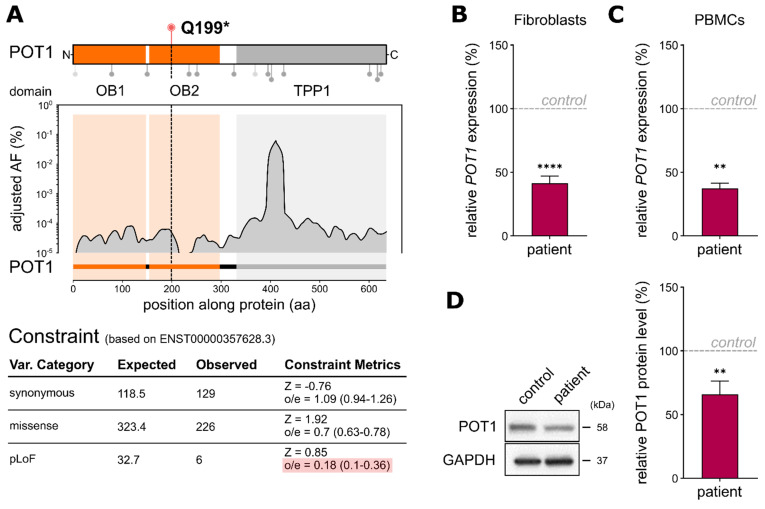
(**A**) Top: POT1 protein structure displaying the interaction domains with single-stranded DNA (1–299) and TPP1 (320–634). Lollipops below depict the positions of variants found in adult AML with predicted loss-of-function variants being displayed in light-grey (adapted from Lim et al., 2021). The variant p.Q199* lies within the interaction domain with single-stranded DNA. Bottom: distribution of the adjusted variant frequencies (AF (%)) along POT1 based on germline variants in the gnomAD noncancer database with the respective constraint metrics (based on Ensembl canonical transcript ENST00000357628.3). (**B**) QRT-PCR analyses of the primary fibroblast samples carrying heterozygous *POT1* p.Q199* compared to the *POT1* WT fibroblasts from an unrelated child. TaqMan probe binds to *POT1* exon 6–7 on chromosome 7. The assays were performed as three independent experiments, each with three technical replicates. (**C**) QRT-PCR analysis of PBMCs of the boy harboring heterozygous *POT1* p.Q199* compared to his father (carrying WT *POT1*). The assays were performed as two independent experiments, each with three technical replicates. (**D**) Representative Western blot and quantitative POT1 protein level analysis of the patient’s fibroblasts harboring *POT1* p.Q199* compared to a control fibroblast sample (WT *POT1*). Three independent Western blots from each genotype were performed. The data represent the means ± SEM. The two-tailed Student’s unpaired *t*-test was performed for the statistical analysis. ** = *p* ≤ 0.01, **** = *p* ≤ 0.0001.

**Figure 3 ijms-22-11572-f003:**
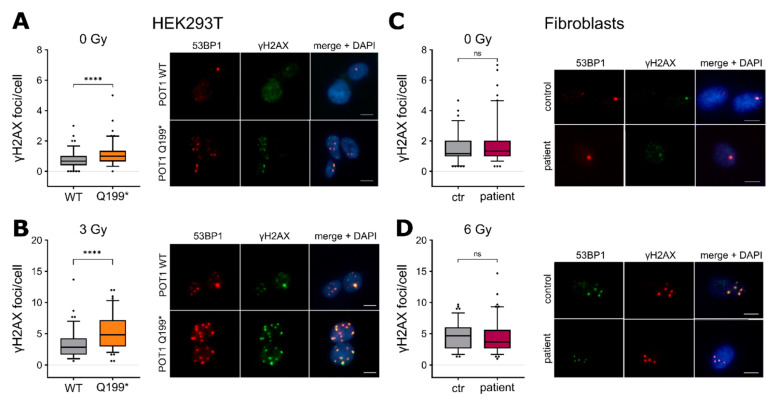
(**A**) Left: Quantification of the yH2AX foci per cell in the HEK293T *POT1* WT and *POT1* p.Q199* cells. Right: Representative images of the yH2AX (green) and 53BP1 (red) foci. DAPI (blue) was used for DNA labeling. Scale bar: 10 µm. (**B**) Left: quantification of yH2AX foci per cell in HEK293T *POT1* WT and *POT1* p.Q199* cells. The cells were exposed to 3 Gy ionising radiation. Right: Representative images analogous to (**A**). (**C**) Immunostaining analogous to (**A**) in the primary fibroblasts of the patient. (**D**) Immunostaining analogous to (**A**) in the primary fibroblasts of the patient with the cells exposed to 6 Gy ionising radiation. The experiments were performed as three biological replicates. The values are expressed in boxplots with whiskers from percentile 5–95. For the statistical analysis, two-tailed Student’s unpaired *t*-test was performed. ns = *p* > 0.05, **** = *p* ≤ 0.0001. ctr = control.

**Figure 4 ijms-22-11572-f004:**
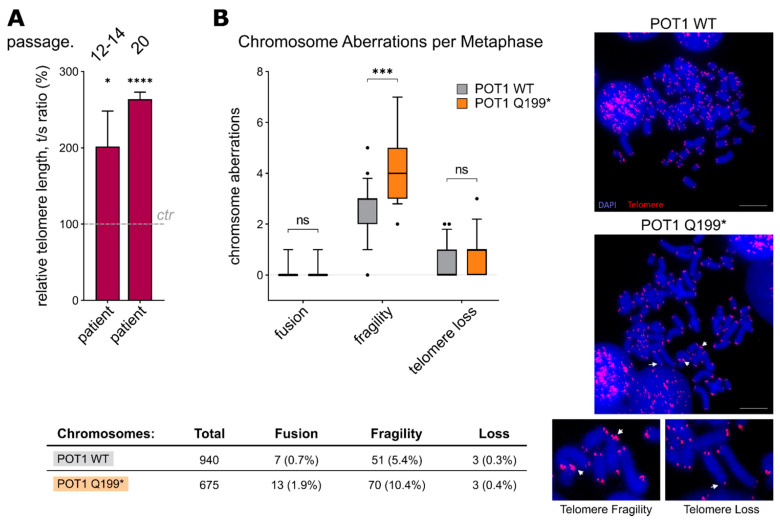
(**A**) Relative telomere length (rTL) analysis by qRT-PCR (comparative ΔΔCt method) with DNA isolated from fibroblast samples (the cells were grown for 12–14 passages or 20 passages). Three biological replicates, each with three technical replicates, were performed. The values are expressed as the means ± SEM. (**B**) Left: Telomere fluorescence in situ hybridisation analysis on metaphase chromosomes of the stably transfected WT and p.Q199* POT1 HEK293T cells. Chromosomal aberrations are categorised in telomere fusion, fragility and loss. Right: Representative images of metaphase chromosomes. Red fluorescence shows telomere signals, and chromosomal DNA was stained with DAPI (blue). White arrows mark the respective chromosomal aberration. Scale bar: 10 µm. ns = *p* > 0.05, * *p =* ≤ 0.05, *** = *p* ≤ 0.001, **** = *p* ≤ 0.0001.

**Figure 5 ijms-22-11572-f005:**
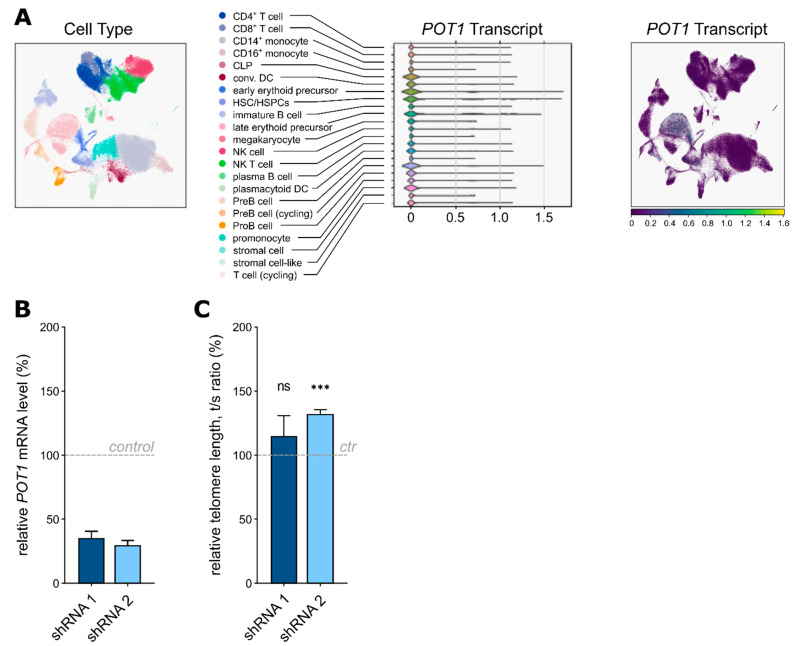
(**A**) Principal component analyses of the single-cell RNA sequencing data displaying POT1 expression within the healthy human bone marrow. CLP: common lymphoid progenitor, DC: dendritic cell, HSC: hematopoietic stem cell, HSPC: hematopoietic stem and progenitor cell, NK: natural killer cell. (**B**) QRT-PCR analysis confirming downregulation of POT1 in the HL-60 cells carrying shRNA 1 and 2 compared to the nontargeting shRNA control. (**C**) Relative telomere length (rTL) analysis by qRT-PCR (comparative ΔΔCt method) with DNA isolated from the HL-60 samples; shRNA 2 showed significant telomere elongation compared to the nontargeting shRNA control (ctr). ns = *p* > 0.05, *** = *p* ≤ 0.001.

## Data Availability

The novel *POT1* variant was submitted to ClinVar (https://www.ncbi.nlm.nih.gov/clinvar/) (accessed on 26 October 2021).

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
