# Peer review of "Germline POT1 Deregulation Can Predispose to Myeloid Malignancies in Childhood"

_ijms, 2021, doi:10.3390/ijms222111572_

Round 1
Reviewer 1 Report
In the current manuscript, the authors repot a germline POT1 mutation which is observed in a case with acute myeloid leukemia. They analyzed the function using in-vitro assays. The experiment is well organized and the manuscript is well written. Here are comments that would make the manuscript clearer.
- What kinds of genomic alterations did the leukemia cells with a POT1 germline mutation have? Also, more detailed diagnosis is required. The data will be helpful to recognize the leukemia development via POT1 deregulation.
- FISH analysis for POT1 will be informative to determine the allele on POT1 in the leukemia cells. Related to this, which region was amplified by qPCR of POT1 in Figures 2B and C?
- What is the status of POT1 in HL-60? Analysis of WT or mutant POT1 overexpression on the cell lines with 7q- will complement the current analysis.
Author Response
Dear Reviewer 1,
Thank you very much for reviewing our manuscript. Please see the attachment for our point-to-point response.
Best regards

Reviewer 2 Report
In this original article the authors describe the germline mutation of the gene POT1 (which is prominently involved in the telomere regulation) in a child with acute myeloid leukemia (AML). In various experiments it is shown that the functional sequelae of this POT1 mutation is a telomeric elongation which might be causative or at least contributory to leukemogenesis.
This paper provides interesting, novel and original data. While the case and the experimental data are well presented, the text might benefit from some detailed editing.
Specific Suggestions for Alterations:
(1) While some 269 pediatric patients have already been screened, it would be of advantage to find more patients with POT1 mutations in order to see whether these cases have the same or different alterations, to determine the true incidence, and other iinteresting data. Have large databases been interrogated (including adults)?
(2) Supplementary Figure 1A: In the forward strand the C > T in the index patient is not clear; maybe these changes can be marked by arrows like in Suppl. Fig. 2A.
(3) Line 94: „derivate chromosome 7“ -> „derivative chromosome 7“.
(4) Supplementary Table 2:
- „deletion 5q, 12p, 20q“ is listed twice;
- „translokation“ -> „translocation“.
(5) Supplementary Table 3: „intrathekal therapy“ -> „intrathecal“.
(6) Line 99: „immunsuppression“ -> „immunosuppression“.
(7) Legend to Figure 1E: „variant carriers“ –> „variant carrier“ (only 1 carrier is shown).
(8) Supplementary Figure 3: Though not so relevant, the y-axis in Fig. 4A and in Suppl. Fig. 3 have different measuring units.
(9) Figure 4B: Title of Fig. 4B (in the figure): „Chromsome Aberrations“ -> „Chromosome Aberrations“.
(10) Line 188: „AML cells lines“ -> „AML cell lines“.
(11) Figure 5A: In the list of different cell types in Fig 5A middle:
- „CD146+ monocytes“?
- abbreviations need to be defined in the legend for: CLP, HSC, HSPC
- „promonocyts“ -> „promonocytes“.
(12) Line 206: „Subsequently“ -> „Accordingly“.
(13) Throughout the text: In correct English, you do not begin a sentence with numbers but with words; for example,
- „one“ instead of „1“ (line 325)
– „three“ instead of „3“ (line 324)
– „twenty“ instead of „20“ (line 338)
– „three“ for „3“ (line 358)
– „one million“ for „1x10e6“ (line 362).
(14) Lines 344, 345 (and possibly elsewhere): „1000“ -> „1,000“.
(15) Line 366: „subjected to slides“ -> „placed on slides“ (first fixed and then onto slides or the other way round?).
(16) Supplementary Figure 4B: Maybe the specific amplified sequences can be highlighted or marked.
Author Response
We would like to thank Referee # 2 for his/her review of our manuscript, his/her positive comments and excellent suggestions. Due to the great eye of Reviewer 2 in spotting errors, we have made multiple corrections to the text/figures and thereby fully addressed comments 2-16 (highlighted in yellow in the revised manuscript file).
Regarding the suggestion to screen additional cohorts (comment 1), we would like to point out the recently published work by Lim et al., (Leukemia, 2021), where 2744 adult patients with hematologic malignancies were analyzed regarding their POT1 status. Within this cohort, 19 unique germline POT1 variants were identified, including three predicted loss of function variants in patients with myeloid malignancies. However, none of these adult patients showed the somatic 7q deletion or disease phenotype that we observed in our pediatric patient.
Nevertheless, we now further extended the search to a cohort of young adults with AML, where we could indeed identify another index case of two brothers with AML (one transplanted, one with a recent relapse). At least one of them harbors a germline POT1 stop gain variant close to the one described here (p.S191*) with concomitant 7q deletion in the tumor (which mimics the somatic phenotype seen in our patient). We are willing to additionally include this family in the here submitted manuscript. Nevertheless, since the initial diagnosis of the first brother was almost 20 years ago, gathering the data, including clinical workup and sequencing verification will take a minimum of 4-6 weeks. Due to the strict revision time of only 8 days, we would kindly ask for an extension in case this adult case should be included.
Best regards
Round 2
Reviewer 1 Report
The authors have responded to the comments from reviewers and have provided additional data. The changes have made the manuscript clearer.